# Combining Bayesian Calibration and Copula Models for Age Estimation

**DOI:** 10.3390/ijerph20021201

**Published:** 2023-01-10

**Authors:** Andrea Faragalli, Edlira Skrami, Andrea Bucci, Rosaria Gesuita, Roberto Cameriere, Flavia Carle, Luigi Ferrante

**Affiliations:** 1Center of Epidemiology, Biostatistics and Medical Information Technology, Università Politecnica delle Marche, 60121 Ancona, Italy; 2Department of Economics, Università degli Studi G. d’Annunzio of Chieti-Pescara, 65127 Pescara, Italy; 3Department of Economics and Law, University of Macerata, 62100 Macerata, Italy; 4AgEstimation Project, University of Macerata, 62100 Macerata, Italy

**Keywords:** age estimation, Bayesian calibration, copula

## Abstract

Accurately estimating and predicting chronological age from some anthropometric characteristics of an individual without an identity document can be crucial in the context of a growing number of forced migrants. In the related literature, the prediction of chronological age mostly relies upon the use of a single predictor, which is usually represented by a dental/skeletal maturity index, or multiple independent ordinal predictor (stage of maturation). This paper is the first attempt to combine a robust method to predict chronological age, such as Bayesian calibration, and the use of multiple continuous indices as predictors. The combination of these two aspects becomes possible due to the implementation of a complex statistical tool as the copula. Comparing the forecasts from our copula-based method with predictions from an independent model and two single predictor models, we showed that the accuracy increased.

## 1. Introduction

Age estimation for living individuals is a common problem in legal medicine. It has a considerable importance in the context of immigration, particularly for undocumented individuals seeking asylum in European countries.

As forced migration continues toward European countries, meaning more unaccompanied young people without documents, the necessity to determine the age of young people has increased. In fact, the assessment of age in children and adolescents is critical for children both to be protected appropriately, and to receive the social and health interventions they need and are entitled to. Therefore, the age estimation should be as accurate as possible. The methods proposed to estimate chronological age from certain maturity indices are mainly based on dental changes or skeletal characteristics of the subjects assessed by radiographs [1,2,3]. In most of the studies devoted to age estimation procedure, the age is estimated using a linear regression model. This approach leads to biased estimates of chronological age [4,5], which is systematically overestimated in younger subjects and underestimated in older ones [6]. As proposed by Lucy and Pollard [5], the use of classical calibration (or “inverse regression”) helps to entirely remove such bias. However, in the “inverse regression”, low correlation between the dependent variable and the covariate may lead to an inaccurate age estimation.

To avoid this issue, Ferrante [7] introduced a Bayesian calibration method with dental maturation as a unique predictor of chronological age. More recently, Bucci [8] followed the same approach using a segmented function to model the relationship between age and dental maturity to consider the changes in the maturation process in juveniles.

A major limitation of the former approaches with Bayesian calibration is that only a single predictor is allowed. However, the use of multiple predictors may be helpful to better predict the chronological age of an individual. Kumagai [9] combined multiple ordinal predictors for age estimation by applying Bayes’ rule to a multivariate continuation ratio model. De Tobel [10] extended this approach, allowing for continuous predictors but assuming a linear relationship between predictors and age. Nevertheless, this methodology relies on the strong assumption of independence between the predictors. We extended the use of the Bayesian calibration approach to multiple continuous predictors, allowing for nonlinear relationships between predictors and age, and combining it with the copula instrument without making any assumption on the dependence structure. The copula function is recognized as a multivariate tool for constructing high flexible joint distribution [11]. For this reason, copulas have been widely used in all fields of research [12,13,14]. This article combines, for the first time, two continuous maturity indices via copula with a Bayesian calibration method and provides an unbiased way to use multiple predictors in the age estimation process. In particular, we used: a dental maturity index, *S* [2], calculated as the sum of normalized open apices in teeth, and a hand–wrist maturation index, *W* [1], defined as the ratio between the total area of the eight carpal bones and the carpal area. The relationship between age and the dental maturity index *S* is inverse and nonlinear: *S* decreases with advanced age in a nonlinear way. On the contrary, the hand–wrist maturation index *W* increases with age.

## 2. Materials and Methods

### 2.1. Sample

A sample of 235 orthopantomographies and wrist X-rays from healthy Italian subjects (114 males and 121 females), where the chronological age of the subjects was known, was used.

The sample included healthy individuals aged 5 to 15 years, with all seven permanent left mandibular teeth, free from malnutrition, wrist fractures, bone diseases, growth and other systemic disorders.

Each patient’s ID number, chronological age and sex were recorded at the beginning of the study. The chronological age of the living subjects was calculated as decimal age by subtracting the date of birth from the date of taking the orthopantomography and wrist X-ray.

The data used in this study were collected and published in a previous study [15] in which ethical approval was obtained.

Ethical review and approval were waived for this study. It was conducted in conformity with the regulations on data management of the Italian law on privacy (Legislation Decree 196/2003 amended by Legislation Decree 101/2018).

### 2.2. Measurements

All orthopantomography and wrist x-ray images were in digital format. The projections were used to determine dental and skeletal maturation, assessed respectively by the sum of open apices of the seven left permanent mandibular teeth (S) [2] and the ratio between the total area of the eight carpal bones and the carpal area for wrist (W) [1]. These methods were chosen because the images were obtained in a simple radiographic position and with a low level of radiation. Two measures, sex and chronological age, were recorded in an Excel file for use as possible predictive variables for age estimation in later statistical analysis.

### 2.3. Statistical Analysis

The relationship between chronological age and the hand–wrist maturation index was almost linear, as shown by Cameriere [1]. For that reason, we assumed that the location parameter of distribution of W followed a simple linear model. On the contrary, the relationship between chronological age and dental maturation, in early stages of life, is not linear (Ferrante [7]) and it can present one or more breakpoints, i.e., points where the relationship changes abruptly. This kind of relationship may be specified in several ways; we modelled S as a segmented function with one breakpoint (Bucci [8]). To this end, the authors investigated a broken-line relationship between S and age (y), following the approach proposed by Muggeo [16].

Consistent with Ferrante [7], a Bayesian calibration method was used (Section A.1). Firstly, we randomly subdivided the dataset into training (about 70% of the total number of observations, *N_tr_* = 160 subjects) and validation (30% of the total number of observations, *N_te_* = 75 subjects) samples, used to develop the models and to measure their predictive performance, respectively. The Bayesian approach to the age estimation problem consists of determining the a posteriori distribution of age, conditioned to the value of the two maturity indices (W, S). In this context, a priori distribution of model parameters and a priori distribution of age are needed: an uninformative uniform prior distribution was used for both, because no a priori information is available.

Considering that *W* and *S* were collected for the same individuals, these two maturity indices could not be assumed to be independent. Every joint distribution function contains information about the marginal behaviour of each variable and their join behaviour (dependence structure). For this reason, to construct the join probability model of *W* and *S*, we referred to copula theory (Section A.2). Copulas enable us to isolate and capture the full structure of dependence in a multivariate distribution, and help with understanding this dependence at a deeper level than simple correlation. The copula theory was introduced by Sklar [11], who proved that for a given set of n random variables, their joint distribution function can be decomposed into the product of marginal distribution functions and a copula. The usefulness of Sklar’s theory stems from the fact that the set of parametric distributions which can be fitted increases substantially, because it is possible to link any n univariate distributions (not necessarily from the same family) with any copula to obtain an unusual but valid n-variate distribution. This allows the researcher to combine any kind of predictors without making any assumptions about their join behaviour. In the literature, several kinds of copula have been implemented. In this work, we focused on three of them: Clayton, Gumbel and Gaussian copulas, that catch respectively lower, upper and no tail dependence (Section A.2). Rotated versions at 90°, 180°, and 270° of Gumbel and Clayton copulas were also considered [14,16,17]. The version of the copula that better describes the dependence structure of our data was selected using the Bayesian Information Criteria (BIC).

According to Sklar’s theory, the bivariate probability model can be characterized by the marginal distributions of S and W and a copula density function.

Within a Bayesian context, we used four different probability models for W and S as follows:$W∼N\left(\mu_{W},\sigma_{W}^{2}\right)$ and $S∼N\left(\mu_{S},\sigma_{S}^{2}\right)$;$W∼N\left(\mu_{W},\sigma_{W}^{2}\right)$ and $S∼ALD\left(\mu_{S},\sigma_{S}^{2},\tau_{S}\right)$;$W∼ALD\left(\mu_{W},\sigma_{W}^{2},\tau_{W}\right)$ and $S∼N\left(\mu_{S},\sigma_{S}^{2}\right)$;$W∼ALD\left(\mu_{W},\sigma_{W}^{2},\tau_{W}\right)$ and $S∼ALD\left(\mu_{S},\sigma_{S}^{2},\tau_{S}\right)$.
where *ALD* is the Asymmetric Laplace Distribution with skew parameters $\tau_{S},\tau_{W}\in \left(0,1\right)$, location parameters $\mu_{W}\left(y,\alpha \right),\mu_{S}\left(y,\beta \right)$ and constant variances $\sigma_{W}^{2}$ and $\sigma_{S}^{2}$.

It should be noted that the joint distribution constructed with the Gaussian copula in the scenario (A) above corresponds to the bivariate normal distribution (Nelsen [18]) of W and S. For each model based on Bayesian calibration, the mode of calibrating distribution was used as the point estimate.

The precision and accuracy of the age estimation models were assessed by:Mean Absolute Error (MAE);Root Mean Squared Error (RMSE);the Inter-Quartile Range of error distribution (IQR*_ERR_*);the mean of the quantile-based 95% Bayesian confidence interval (MCI_95%_) of the calibrating distribution.

## 3. Results

The methodology here presented was applied to a dataset of 235 orthopantomographies and wrist X-rays from healthy Italian subjects (114 males and 121 females), where the chronological age of the subjects was known (in years). In this sample, y ranged between 5 and 15 years, S between 0.04 and 4.51, and W between 0.27 and 0.96. We verified that there was no difference in the distribution of age between the training and validation samples by performing a Kolmogorov–Smirnov test, which did not reject the null hypothesis of no differences between the two distributions. Figure 1 shows the relationship between age and the predictors in both the training and validation samples. The estimated segmented parameter was ψ=11.31 years. This means that, in our training sample, the dental maturation slope abruptly changed in proximity to this value. According to the Bayesian Information Criteria (BIC), the most suitable versions of the Gumbel and Clayton copula selected were the Rotated Gumbel (270°) and the Rotated Clayton (90°) (Table 1), both able to capture an asymmetric dependency structure between W and S (a high value of W corresponds to a low value of S). The analysis of the predictive performance of the considered models is reported in Table 2.

The skewness parameters of the marginal distributions were τ=0.43 for S and τ=0.51 for W, showing that the distribution of S was quite asymmetric while the distribution of W could be considered symmetric. The results in Table 2 show the good predictive performance of the copula instrument with a different distribution for age predictors. In fact, the presented copula models outperformed the two single-predictor models and the independent one in each of the measures considered. None of the copula models showed any significant bias in the estimated residuals on chronological age. Comparing the outcomes among panels, it may be noticed that they are mostly comparable across the copula models, while the assumption of independent age predictors seems not to provide accurate estimates of chronological age. The models with a single predictor exhibited less accurate predictions, underlining that the use of a larger set of anthropometric characteristics of the subject can lead to a better estimate of the chronological age. Figure 2 reports the calibration distributions constructed using the Rotated Clayton copula model with S for different values of S and W. This helps us to understand how distribution of age varies in relation to the anthropometric indices and highlights two relevant aspects: age distribution is non-normal and exhibits heteroskedasticity.

## 4. Discussion

Accurately estimating chronological age assumes a crucial role in forensic science and legal medicine for solving a variety of legal issues concerning criminal liability, majority status and the identification of both living and dead individuals. We introduced a new method that permits the practitioner to use more than one chronological age predictor and completely remove the bias that exists when linear regression is used. To the best of our knowledge, this is the first study that bases the age estimation process on two different anthropometric measures allowing for nonlinear relations between age and predictors without making any assumptions about the dependence structure between predictors. De Tobel [10] highlighted the necessity of an appropriate statistical approach for handling the dependence between predictors. Our study seeks to capture the joint dependence of two maturity indices, the sum of open apices in teeth, and the ratio between the total area of the eight carpal bones and the carpal area, via the combination of the flexible copula function and the Bayesian calibration. Although implemented fairly widely in other fields, copulas have not yet been used in the process of estimating chronological age. This approach is the most flexible way to combine multiple predictors, and the only one producing unbiased estimates.

In our application on real data, the copula models outperformed the models with a single predictor and the model in which the age predictors were assumed to be independent. In fact, the copula models appeared to be more accurate and robust than the independent assumption model when multiple predictors are taken into account, because they exhibited the lowest MAE, RMSE, IQR*_ERR_*and MCI_95%_. The results obtained in our study also highlighted that the performance of the approach here presented was not affected by the choice of the marginal distribution, meaning that it predicts with greater accurately than the more simplistic hypothesis of independence, regardless of the distribution assumptions made on the involved variables.

Our study found that the best model for estimating chronological age had an average accuracy of ±1.9 years, using a combination of dental and wrist bone maturation as indicators of development. This level of accuracy is similar to or better than that of other Bayesian models that have been developed for this purpose. For example, Rynkiewicz et al. [19] reported an average accuracy of ±3.5 years using a wrist bone maturation method, while Chen et al. [20] reported an average accuracy of ±3.7 years using a combination of dental and wrist bone maturation. Finally, this paper extends to two indices the methodology proposed in Bucci et al. [8], where the best-performing method exhibited an accuracy of ±2.8 years. These findings suggest that the model developed in our study is a promising tool for accurately estimating chronological age in children and adolescents.

A limitation of this study is that the findings may be not representative of different populations, and the specific characteristics of the sample may have influenced the main conclusions. However, the general results from our analysis are consistent throughout the simulation with different settings and the empirical application (as described in Section A.3), which suggests that the model can be easily generalized to populations with different characteristics.

Furthermore, it should be mentioned that it may not be common to obtain both tooth and wrist maturity indices in practice. However, if this information is available, both indices should be used in combination to achieve a higher level of accuracy in age estimation.

## 5. Conclusions

In conclusion, the use of a larger set of anthropometric characteristics of the subject can lead to a better estimate of the subject’s chronological age than using a single measure.

Possible future developments relate to the use of additional individual characteristics as predictors, such as sex or other anthropometric measurements, which could lead to an improved accuracy in age estimation. Notwithstanding that the use of more than two predictors would imply an augmented computational effort, the whole procedure could be easily applied in such a scenario by using the Pair Copula Construction technique.

## Figures and Tables

**Figure 1 ijerph-20-01201-f001:**
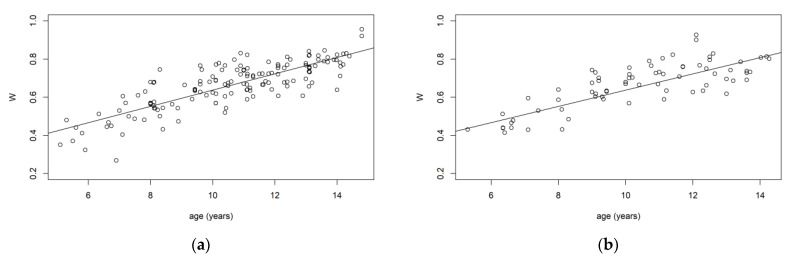
Relationship between age and the maturity indices, *S* and *W*, in the training and validation samples. (**a**) Relationship between age and W in training sample. (**b**) Relationship between age and W in validation sample. (**c**) Relationship between age and S in training sample. (**d**) Relationship between age and S in validation sample.

**Figure 2 ijerph-20-01201-f002:**
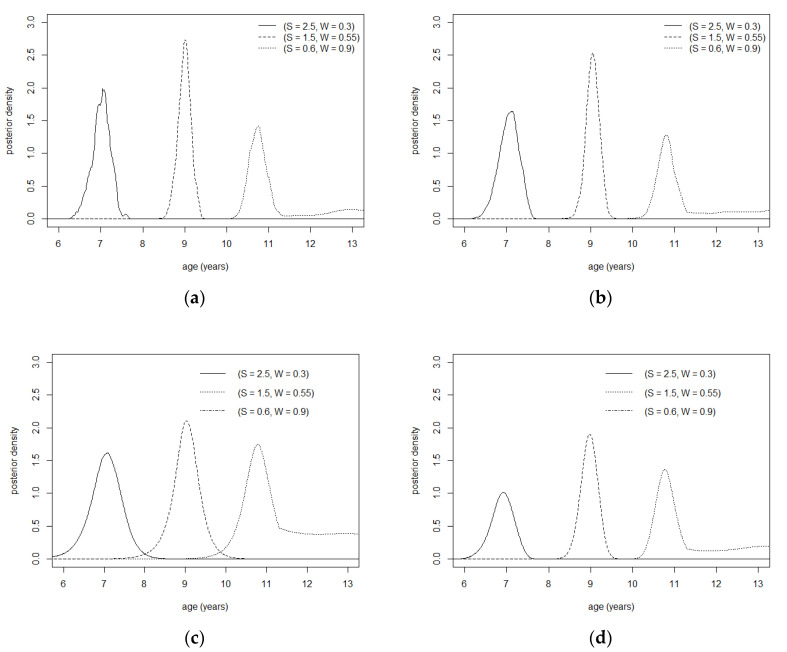
Posterior age distribution for given values of *S* and *W* with a Rotated Clayton (90°) model in the validation sample. (**a**) *S* and *W* normally distributed. (**b**) *S* and *W* asymmetric Laplace distributed. (**c**) *S* asymmetric Laplace distributed and *W* normally distributed. (**d**) *S* normally distributed and *W* asymmetric Laplace distributed.

**Table 1 ijerph-20-01201-t001:** Type of Clayton and Gumbel copula selection.

Copula	BIC
Clayton:	
Not rotated Clayton	−40.879
R-Clayton C. (90°)	−119.814
R-Clayton C. (180°)	−40.992
R-Clayton C. (270°)	−56.342
Gumbel:	
Not rotated Gumbel	−42.033
R-Gumbel C. (90°)	−79.181
R-Gumbel C. (180°)	−42.168
R-Gumbel C. (270°)	−124.289

Abbreviations: BIC, Bayesian Information Criteria; R, Rotated.

**Table 2 ijerph-20-01201-t002:** Prediction comparison of the considered models with observed data (validation sample).

Model	MAE	RMSE	IQR*_ERR_*	MCI_95%_
**Panel A: S~N; W~N**				
Segmented (*S* single predictor)	1.13	1.41	(−1.46; 0.35)	2.62
Linear (*W* single predictor)	1.21	1.68	(−1.44; 0.67)	2.81
Independent	1.11	1.40	(−1.42; 0.28)	2.17
Gaussian C.	1.03	1.32	(−1.18; 0.65)	1.92
R-Gumbel C. (270°)	1.02	1.32	(−1.16; 0.63)	1.94
R-Clayton C. (90°)	1.01	1.31	(−1.16; 0.63)	1.96
**Panel B: S~ALD (τ = 0:43); W~N**				
Segmented (*S* single predictor)	1.09	1.39	(−1.46; 0.35)	2.33
Linear (*W* single predictor)	1.21	1.68	(−1.44; 0.67)	2.81
Independent	1.01	1.34	(−1.17; 0.59)	2.07
Gaussian C.	1.02	1.32	(−1.16; 0.61)	1.99
R-Gumbel C. (270°)	1.03	1.31	(−1.13; 0.65)	2.41
R-Clayton C. (90°)	1.01	1.30	(−1.14; 0.62)	2.57
**Panel C: S~N; W~ALD (τ = 0:51)**				
Segmented (*S* single predictor)	1.13	1.41	(−1.46; 0.35)	2.62
Linear (*W* single predictor)	1.20	1.65	(−1.41; 0.66)	2.78
Independent	1.28	1.55	(−0.78; 1.16)	8.51
Gaussian C.	1.03	1.32	(−1.10; 0.74)	2.26
R-Gumbel C. (270°)	1.02	1.32	(−1.15; 0.61)	1.98
R-Clayton C. (90°)	1.01	1.31	(−1.23; 0.57)	2.06
**Panel D: S~ALD (τ = 0:43); W~ALD (τ = 0:51)**				
Segmented (*S* single predictor)	1.09	1.39	(−1.46; 0.35)	2.33
Linear (*W* single predictor)	1.20	1.65	(−1.41; 0.66)	2.78
Independent	1.12	1.41	(−1.51; 0.18)	3.12
Gaussian C.	1.03	1.32	(−1.10; 0.74)	1.91
R-Gumbel C. (270°)	1.02	1.30	(−1.10; 0.74)	1.93
R-Clayton C. (90°)	1.03	1.31	(−1.11; 0.73)	1.95

Abbreviations: MAE, mean absolute error; RMSE, root mean squared error; IQR*_ERR_*, interquartile range of error; MCI_95%_, mean of 95% Bayesian confidence interval; C, Copula; R, Rotated.

## Data Availability

Software in the form of R code, together with a sample input data set and complete documentation is available on request from the corresponding author (r.gesuita@staff.univpm.it).

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
