# Peer review of "Combining Bayesian Calibration and Copula Models for Age Estimation"

_ijerph, 2023, doi:10.3390/ijerph20021201_

Round 1

Reviewer 1 Report

The authors present a unique study that provides valuable information on age estimation based on anthropometric characteristics of sub-adults aged 5-15 years. A detailed statistical analysis was performed, which showed a good accuracy of the developed prediction model. Despite all efforts, there are some aspects that should be revised and completed before publication. Please refer to the comments and recommendations below:

-      In the Material and Method section – Was the chronological age calculated as a decimal age from the date of birth and the exact date of the OPG examination?

- Be more specific on inclusion/exclusion criteria for teeth - what about extracted teeth? Or impacted teeth, how were these evaluated?

-      What exactly do you mean by "free of missing teeth"? Exactly which teeth were studied?

-      Since this is a retrospective study, at least mention the ethical approval of the institution.

-      Provide more specific inclusion/exclusion criteria for the wrist evaluation - fractures, bone diseases in sub-adults or other abnormalities that could introduce bias.

-      As far as I know, continuous and independent variables are conditional in Bayesian analysis, as suggested by De Tobel 2020. In your study, you use both continuous variables, but not the independent ones, as you mention in the manuscript. So, if the maturity index has a nonlinearity (older individuals and lower index value), why not use the developmental stages of the teeth, even if it is not a continuous variable, but they are linear (higher stage, older individual)? This should be mentioned in the introduction and discussion section.

-      Use a training sample and a valid sample instead of train/test.

-      The discussion section is completely missing. I understand that this is a pilot study that brings new knowledge, but at least a comparison of the accuracy of the present model with previously published models should be mentioned. The population specificity of the model, if any, should also be discussed. Is it necessary to perform such studies on other populations?

-      From a mathematical-statistical point of view, I understand that you want to minimize bias and therefore suggested using two predictors for chronological age (lines 214-215 in the Conclusions section). However, in practice, this could sometimes be counterproductive, as it might be difficult to obtain even one predictor (either tooth or bone maturation assessment). So, your work definitely provides valuable insights presented in an organized and comprehensive manner, but on the other hand, this point should not be omitted

Reviewer 2 Report

The present manuscript does not fall within the scope of this journal (International Journal of Environmental Research and Public Health). Instead, it should be submitted to the journal titled "Forensic Sciences" (https://www.mdpi.com/journal/forensicsci).

Author Response

Thank you for the suggestion, but we intend to continue with this journal by responding to the comments of two other reviewers

Reviewer 3 Report

I congratulate the authors for bringing out this novel study. I believe, the combined Bayesian Calibration and Copula models as demonstrated by the example would be helpful in age estimation in forensic casework. The manuscript is well structured and presented. However, the authors need to include an ethical statement and consent for the study in the methodology section. Furthermore, the conclusion section summarizes the study findings and states whether the study's objectives were met. Therefore, I suggest the authors did not cite any references in the conclusion section.

Author Response

Pleese see the attachment

Round 2

Reviewer 1 Report

The authors have incorporated all the suggestions in the revised version, hence acceptable for publication.